

# Unraveling the dynamics of Xanthomonas' flagella: insights into host-pathogen interactions

Maria L. Malvino

Crop Sciences, University of Illinois at Urbana-Champaign, Urbana, Illinois, United States

## ABSTRACT

Understanding the intricate interplay between plants and bacteria is paramount for elucidating mechanisms of immunity and disease. This review synthesizes current knowledge on the role of flagella in bacterial motility and host recognition, shedding light on the molecular mechanisms underlying plant immunity and bacterial pathogenicity. We delve into the sophisticated signaling network of plants, highlighting the pivotal role of pattern recognition receptors (PRRs) in detecting conserved molecular patterns known as microbe-associated molecular patterns (MAMPs), with a particular focus on flagellin as a key MAMP. Additionally, we explore recent discoveries of solanaceous-specific receptors, such as FLAGELLIN SENSING 3 (FLS3), and their implications for plant defense responses. Furthermore, we examine the role of bacterial motility in host colonization and infection, emphasizing the multifaceted relationship between flagella-mediated chemotaxis and bacterial virulence. Through a comprehensive analysis of flagellin polymorphisms within the genus Xanthomonas, we elucidate their potential impact on host recognition and bacterial pathogenicity, offering insights into strategies for developing disease-resistant crops. This review is intended for professionals within the fields of crops sciences and microbiology.

## INTRODUCTION

The intricate dance between plants and bacteria represents a constant battle for survival, shaped by millennia of coevolutionary dynamics. At the heart of this complex interrelation lies the recognition of conserved molecular patterns known as microbe-associated molecular patterns (MAMPs) by pattern recognition receptors (PRRs) on the host cell membrane (*Saijo, Loo & Yasuda, 2018*; *Zipfel et al., 2004*). This fundamental mechanism serves as the cornerstone of the plant immune system, orchestrating a rapid and robust defense response upon detection of potential microbial invaders. Among the diverse array of MAMPs, flagellin, a structural component of bacterial flagella, stands out as a potent elicitor of immune responses in plants.

Studies over the past decades have illuminated the pivotal role of flagellin in initiating pattern-triggered immunity (PTI), the first line of defense mounted by plants against invading pathogens. The landmark discovery of the pattern recognition receptor

Corresponding author
Maria L. Malvino,
lmalvino@yahoo.com

FLAGELLIN SENSITIVE 2 (FLS2) in *Arabidopsis thaliana* marked a significant breakthrough in our understanding of flagellin perception in plants (*Gómez-Gómez & Boller, 2000*; *Zipfel et al., 2004*). FLS2, a leucine-rich repeat transmembrane receptor, serves as the sentinel for detecting the presence of flagellin in the plant environment, triggering a cascade of immune responses, including the generation of reactive oxygen species (ROS), modulation of host gene transcription, and activation of mitogen-activated protein kinase (MAPK) cascades (*Asai et al., 2002*; *Jones & Dangl, 2006*; *Torres, Dangl & Jones, 2002*; *Zhang & Klessig, 2001*).

Building upon this foundational discovery, recent studies have unveiled the existence of solanaceous-specific receptors, such as FLAGELLIN SENSING 3 (FLS3), which recognize distinct epitopes of flagellin and contribute to plant defense responses (*Cai et al., 2011*; *Hind et al., 2016*). The identification of FLS3 in tomato (*Solanum lycopersicum*), potato (*Solanum tuberosum*), and pepper (*Capsicum annuum*) plants has broadened our understanding of the diversity of plant immune recognition systems and underscored the complexity of host-pathogen interactions (*Roberts et al., 2020*).

In parallel, research efforts have elucidated the pivotal role of bacterial motility, facilitated by flagella, in mediating host colonization and infection. Bacteria employ flagella to navigate gradients of chemoattractants, enabling efficient recognition and colonization of host plants (*Scharf, Hynes & Alexandre, 2016*). The convoluted interplay between flagellar-mediated chemotaxis and bacterial virulence highlights the multifaceted nature of host-pathogen interactions, with motility often serving as a determinant of pathogen fitness and pathogenicity (*Josenhans & Suerbaum, 2002*; *Ottemann & Miller, 1997*).

Drawing upon insights from these diverse fields of research, this review aims to provide a comprehensive overview of the dynamics of flagella and bacteria in the context of host-pathogen interactions. We delve into the molecular mechanisms governing flagellin recognition in plants, explore the role of bacterial motility in host colonization and infection, and examine the implications of flagellin polymorphisms in bacterial pathogenicity, with a particular focus on the genus *Xanthomonas*. By synthesizing current knowledge and highlighting key research findings, it is aimed to shed light on the labyrinthine dynamics between flagella, bacteria, and host plants, offering insights into strategies for enhancing plant immunity and mitigating agricultural losses.

## SURVEY METHODOLOGY

Various methods are utilized to search for relevant articles. One common approach involves using keywords on popular platforms such as Google Scholar. Keywords like "*Xanthomonas*," "flagella," "FLS2," "FLS3," "pathogenicity," "MAMP," and "PRR" are commonly used to refine the search and focus on specific areas of interest. The search is further refined by adjusting the keywords to tailor the search results to the desired topic. Only articles written in English are considered.

Once a substantial number of articles are gathered, their abstracts are reviewed to determine their suitability for inclusion in the review process. Articles that significantly contribute to the research subject are selected for further review.

Additionally, cross-references are explored by examining papers with notable research contributions to identify additional relevant sources.

## Flagella and bacterial motility

Bacterial motility, facilitated by flagella, represents a fundamental strategy for bacterial colonization, dissemination, and infection in the plant environment. Flagella, intricate whip-like appendages protruding from bacterial cells, serve as molecular propellers, enabling bacteria to navigate through complex environmental gradients with remarkable precision (*Berg, 2003*). The dynamic rotation of flagella propels bacteria through diverse habitats, ranging from liquid media to solid surfaces, facilitating their exploration and exploitation of ecological niches (*Berg & Brown, 1972*).

Chemotaxis, the directed movement of bacteria in response to chemical gradients, plays a pivotal role in guiding bacterial motility towards favorable environments (*Wadhams & Armitage, 2004*). Bacteria possess sophisticated chemosensory systems that allow them to detect and respond to a wide range of chemical cues present in their surroundings. Chemoattractants, such as nutrients, amino acids, and organic acids, serve as signals for bacterial navigation, guiding them towards nutrient-rich microenvironments conducive to growth and proliferation (*Hazelbauer, Falke & Parkinson, 2008*).

Flagellar-mediated chemotaxis enables bacteria to locate and colonize host plants, exploiting nutrient-rich niches provided by root exudates and leaf surfaces (*Scharf, Hynes & Alexandre, 2016*). Soil-borne bacteria, such as rhizobia and plant growth-promoting rhizobacteria (PGPR), utilize flagellar motility to navigate towards plant roots, initiating symbiotic interactions or establishing beneficial associations with host plants (*Bashan, Holguin & De-Bashan, 2004*). Additionally, foliar pathogens, including members of the genus *Xanthomonas*, employ flagella to move across leaf surfaces, facilitating the initial stages of host colonization and infection (*Ottemann & Miller, 1997*).

Once bacteria successfully reach their plant hosts, they employ a variety of strategies to infect and colonize the plant tissues. Foliar pathogens like *Xanthomonas* species use their flagella not only for movement but also to penetrate plant surfaces and establish infection. This involves the secretion of effector proteins *via* a type III secretion system, which suppress plant immune responses and facilitate bacterial invasion (*Büttner & Bonas, 2010*). Root-associated bacteria like *Pseudomonas fluorescens* and *Pseudomonas syringae* utilize similar mechanisms to infect root tissues, leading to diseases such as root rot and vascular wilt (*Preston, 2004*).

The ability of bacteria to switch between different modes of motility, such as swimming and swarming, further enhances their adaptability to diverse environmental conditions (*Kearns, 2010*). Swimming motility, characterized by flagellar-driven movement in liquid media, allows bacteria to explore and colonize aqueous habitats, whereas swarming motility enables rapid surface colonization through coordinated flagellar movement on semi-solid surfaces (*Kearns & Losick, 2003*). Swarming motility, in particular, has been implicated in the early stages of bacterial infection, facilitating the efficient dissemination of pathogens across host surfaces (*Kearns, 2010*).

Flagellar motility also plays a critical role in biofilm formation, a complex process involving the coordinated attachment, aggregation, and maturation of bacterial communities on biotic or abiotic surfaces (*Costerton, Stewart & Greenberg, 1999*). Bacteria utilize flagella to initiate surface attachment and explore microenvironments conducive to biofilm formation, eventually transitioning to sessile lifestyles characterized by matrix production and community development (*Kearns, 2010*). The formation of biofilms enhances bacterial survival and persistence in the environment, providing protection against environmental stresses and antimicrobial agents (*Hall-Stoodley, Costerton & Stoodley, 2004*).

Infection establishment and persistence are often facilitated by biofilm formation, as biofilms protect bacterial communities from host immune responses and increase their resistance to antimicrobial treatments. For instance, biofilms formed by *Pseudomonas aeruginosa* on plant surfaces can resist phagocytosis by plant immune cells, thereby ensuring prolonged colonization and infection (*Hogan & Kolter, 2002*).

Recent studies have advanced our understanding of the interaction mechanisms between flagellin and bacterial migration. *Nakamura & Minamino (2023)* elucidated how the bacterial flagellum drives motility by functioning as a molecular screw. Their study focused on the structural and dynamic aspects of the flagellar motor, crucial for bacterial migration and chemotaxis, highlighting the conversion of ion flux into mechanical work for motor rotation. Specifically, this supramolecular assembly includes three distinct functional components: a basal body that functions as a bidirectional rotary motor with several force generators, each serving as a transmembrane proton channel to link proton flow through the channel with torque generation; a filament acting as a helical propeller to create propulsion; and a hook that operates as a universal joint, transmitting the torque generated by the rotary motor to the helical propeller (*Minamino & Kinoshita, 2023*).

*Singh et al. (2024)* dived into the details of how the motor transmits torque and direction to the flagellar rod, which is key in further elucidating bidirectional rotation during motility and chemotaxis. They provided detailed insights into the structural mechanisms by which the bacterial flagellum rotates and switches direction. Using cryogenic electron microscopy (CryoEM), the researchers captured high-resolution structures of the flagellar motor's inner membrane MS-ring and the cytoplasmic C-ring in both counterclockwise and clockwise orientations. They observed significant conformational differences between these states, including a 180° shift in key protein domains that reposition\binding sites for the MotA/B complexes, which are involved in torque generation. The study also highlighted how specific regulatory proteins interact with these conformations to switch the motor's rotational direction.

In the study by *Matilla & Krell (2023)*, the authors explore targeting bacterial motility and chemotaxis as a novel strategy to combat bacterial pathogens, offering an alternative approach as traditional antibiotics encounter growing resistance. They review various compounds that inhibit chemotaxis by interfering with signal recognition or flagellar rotation, highlighting how these mechanisms reduce bacterial virulence by preventing effective host invasion and tissue attachment. While this approach doesn't affect all pathogens equally and may not completely abolish virulence, it offers a promising

alternative that doesn't promote resistance. The study discusses the pros and cons, noting that effective inhibitors with low IC50 values are available, but more research is needed to explore additional targets within the chemotactic apparatus for greater efficacy. This strategy aims to reduce infection by disrupting disease-causing mechanisms rather than bacterial growth, thus minimizing the development of resistant strains.

*Terashima, Kojima & Homma (2008)* examined the structural variations of flagellin and their impact on bacterial motility, revealing how modifications in flagellin structure can influence bacterial swimming and swarming behaviors. These studies collectively underscore the complex relationship between flagellar structure, function, and bacterial motility, offering new perspectives on how bacteria adapt their motility mechanisms to diverse environmental conditions and host interactions.

In summary, flagella-mediated bacterial motility represents a multifaceted adaptation that enables bacteria to navigate through diverse environments, locate host plants, and initiate interactions leading to colonization and infection. The dynamic interplay between flagellar-driven chemotaxis, motility switching, and biofilm formation underscores the versatility and resilience of bacterial pathogens in the face of host defenses.

## Flagellin perception in plants

Plants recognize flagellin as a critical component of the innate immune system, enabling rapid and effective defense responses against invading pathogens. Flagellin, a highly conserved proteinaceous component of bacterial flagella, serves as a potent elicitor of immune responses in diverse plant species, triggering the activation of pattern-triggered immunity (PTI) upon detection by pattern recognition receptors (PRRs) on the host cell surface (*Gómez-Gómez & Boller, 2000*).

The *Arabidopsis thaliana* receptor kinase FLAGELLIN SENSITIVE 2 (FLS2) was the first PRR identified to perceive bacterial flagellin and initiate immune signaling cascades in plants (*Gómez-Gómez & Boller, 2000*; *Zipfel et al., 2004*). FLS2, a transmembrane receptor with extracellular leucine-rich repeat (LRR) domains, specifically recognizes the highly conserved N-terminal epitope of flagellin, known as flg22, leading to the activation of defense responses, including the production of reactive oxygen species (ROS), activation of mitogen-activated protein kinase (MAPK) cascades, and transcriptional reprogramming of defense-related genes (*Asai et al., 2002*; *Chinchilla et al., 2006*; *Felix et al., 1999*; *Gómez-Gómez, Felix & Boller, 1999*; *Zipfel et al., 2004*).

In addition, the TLR5 receptor in animals recognizes flagellin, demonstrating a similar mechanism where highly conserved microbial features are detected to trigger an immune response. This highlights the evolutionary conservation of flagellin recognition mechanisms across different kingdoms of life. Unlike FLS2 in plants, which recognizes the conserved N-terminal epitope of flagellin known as flg22, TLR5 detects a different region of flagellin, specifically the D1 domain. This distinction in recognition sites underscores the diverse evolutionary strategies employed by different organisms to detect and respond to bacterial flagellin (*Smith et al., 2003*; *Andersen-Nissen et al., 2005*).

For example, in *Arabidopsis*, flg22 perception leads to the rapid production of ROS, which acts as a signaling molecule to reinforce cell walls and activate downstream defense

genes. This oxidative burst is a hallmark of PTI and plays a vital role in limiting pathogen spread. Additionally, the activation of MAPK cascades by FLS2 results in the phosphorylation of transcription factors that regulate the expression of defense genes involved in antimicrobial production and other defensive measures (*Asai et al., 2002*).

The perception of flagellin by FLS2 represents a hallmark of plant immune signaling, illustrating the ability of plants to detect and respond to conserved microbial features through specific recognition mechanisms (*Chinchilla et al., 2006*; *Gómez-Gómez, Felix & Boller, 1999*; *Zipfel et al., 2004*). The importance of flagellin perception in plant immunity is underscored by the broad conservation of FLS2 homologs across diverse plant species, highlighting the evolutionary significance of flagellin recognition in shaping host-pathogen interactions (*Robatzek & Wirthmueller, 2013*).

In rice, for instance, the OsFLS2 receptor recognizes flagellin and triggers similar immune responses to those observed in *Arabidopsis*, including ROS production and MAPK activation, thereby providing protection against bacterial pathogens like *Xanthomonas oryzae* pv. *oryzae*, the causative agent of bacterial blight (*Wang et al., 2015*).

Furthermore, the tertiary structure of flagellin and its interaction with the plant receptor FLS2 is critical in understanding bacterial motility and immune evasion. Flagellin forms a complex with FLS2, a pattern recognition receptor in plants, triggering immune responses (*Chinchilla et al., 2007*). This interaction is highly specific and involves precise binding of conserved regions of flagellin to the extracellular leucine-rich repeat (LRR) domain of FLS2. Recent advances have elucidated the detailed tertiary structures of flagellin and FLS2, providing insights into the molecular basis of this interaction.

The tertiary structure of FLS2 comprises an extensive LRR domain that forms a solenoid structure, facilitating the specific recognition and binding of flagellin. This LRR domain is crucial for the high-affinity binding to conserved epitopes of flagellin, particularly the flg22 peptide, a highly conserved 22-amino acid sequence within flagellin. The FLS2 receptor also features a transmembrane domain that anchors it to the plant cell membrane and a cytoplasmic kinase domain responsible for initiating downstream signaling cascades upon flagellin binding (*Robatzek & Wirthmueller, 2013*).

Recent structural studies using techniques such as cryo-electron microscopy (cryo-EM) and X-ray crystallography have provided high-resolution images of the flagellin-FLS2 complex, revealing the sophisticated details of their interaction. These studies have shown that the LRR domain of FLS2 undergoes conformational changes upon binding to flagellin, enhancing the stability of the complex and promoting signal transduction (*Wang & Chai, 2020*).

The structural variations in flagellin, as discussed by *Saijo, Loo & Yasuda (2018)*, can significantly affect its recognition by FLS2, thereby influencing bacterial motility and pathogenicity. For instance, certain mutations in the flagellin structure can either enhance or diminish its binding affinity to FLS2, affecting the efficiency of immune recognition and the subsequent immune response. Understanding these structural nuances is crucial for developing strategies to manipulate plant immunity and improve resistance against bacterial pathogens.

In addition to FLS2, recent studies have identified solanaceous-specific receptors, such as FLAGELLIN SENSING 3 (FLS3), which recognize distinct epitopes of flagellin, flgII-28, and contribute to plant defense responses (*Cai et al., 2011*; *Hind et al., 2016*). FLS3, initially characterized in tomato (*Solanum lycopersicum*), potato (*Solanum tuberosum*), and pepper (*Capsicum annuum*), represents a novel layer of flagellin perception in solanaceous plants, expanding the repertoire of plant immune receptors and diversifying the mechanisms of host defense (*Hind et al., 2016*; *Roberts et al., 2020*) (Fig. 1).

The identification of FLS3 highlights the complexity of flagellin perception in plants and raises intriguing questions about the evolutionary origins and functional diversity of flagellin receptors across different plant species. Further research into the molecular mechanisms governing flagellin recognition by FLS3 and its downstream signaling pathways promises to provide new insights into the dynamics of plant immunity and host-pathogen coevolution.

## Flagellin polymorphisms and bacterial pathogenicity

The genus *Xanthomonas* encompasses a diverse group of plant-pathogenic bacteria that cause diseases in a wide range of economically important crops, including rice, citrus, tomato, and pepper. Among the numerous virulence factors employed by xanthomonads, flagella play a crucial role in mediating host colonization, biofilm formation, and disease development (*Ryan et al., 2011*).

Flagellin, encoded by the *fliC* gene, represents a major structural component of *Xanthomonas* flagella and serves as a potent elicitor of plant immune responses. The recognition of *Xanthomonas* flagellin by host plants triggers the activation of pattern-triggered immunity (PTI), leading to the induction of defense responses and restriction of bacterial growth (*Sun et al., 2006*). Consequently, *Xanthomonas* spp. have evolved sophisticated mechanisms to evade or suppress PTI, enabling successful colonization and infection of host plants (*Newman et al., 2013*).

One strategy employed by *Xanthomonas* spp. to evade host detection involves the sophisticated manipulation of flagellin structure through amino acid substitutions or post-translational modifications. Flagellin, as a highly conserved protein across bacterial species, serves as a primary target for host immune recognition. By altering specific amino acid residues or undergoing modifications such as glycosylation or acetylation, *Xanthomonas* bacteria can effectively disguise or modify flagellin's molecular signature, thereby evading or attenuating the host's immune response (*Burdman et al., 2004*; *da Silva et al., 2002*; *Gottig et al., 2009*).

These modifications allow *Xanthomonas* spp. to fine-tune their interactions with host plants, influencing factors such as host range, pathogenicity, and virulence. For instance, certain amino acid substitutions may confer a selective advantage by enhancing the affinity of flagellin for specific plant receptors, thereby promoting successful colonization and infection (*Sun et al., 2006*). Additionally, post-translational modifications can modulate flagellar motility and adherence properties, facilitating bacterial attachment to host surfaces and biofilm formation, crucial steps in the establishment of infection (*Burdman et al., 2004*; *da Silva et al., 2002*; *Gottig et al., 2009*).

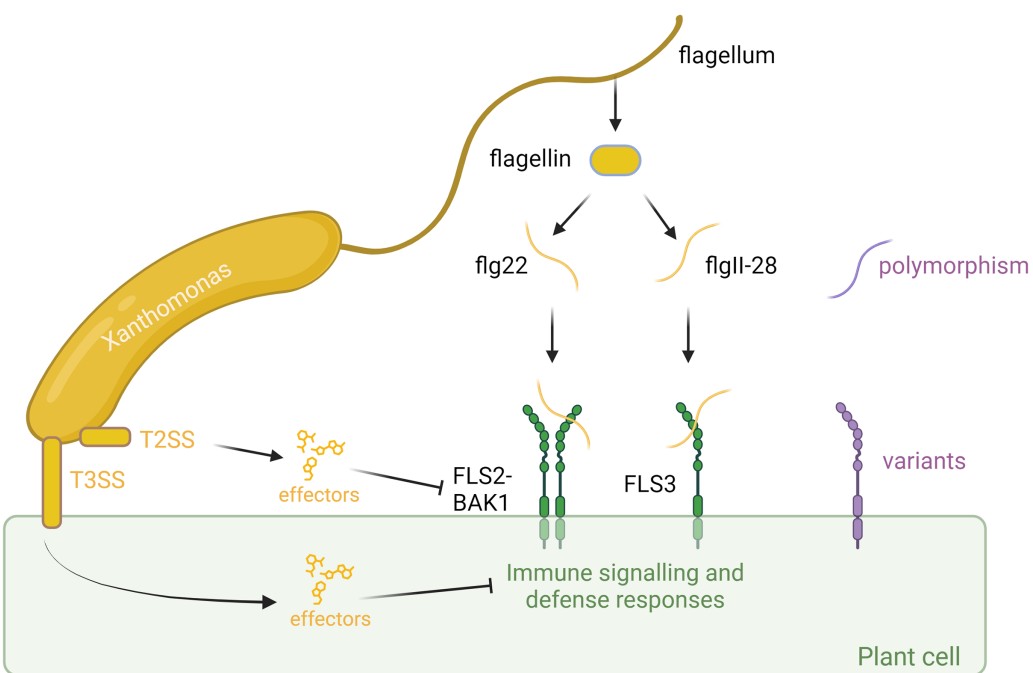

**Figure 1 Plant-associated bacteria employ various strategies to counteract pathogen-associated molecular pattern (PAMP)-triggered immunity (PTI) initiated by flagellin detection in plants.** Plants identify bacteria by recognizing conserved molecular patterns, such as peptides derived from the flagellin protein. This recognition is facilitated by the receptors FLS2 or FLS3, along with the coreceptor BAK1, which forms a complex with the flagellin-derived elicitor peptides. The polymorphism in flagella enables bacteria to evade detection or diminish the effectiveness of plant immune responses. However, plants can counter this by evolving receptor variants that are capable of recognizing these new flagellar polymorphisms, thereby restoring their ability to detect and respond to bacterial invaders. In addition to flagella, bacteria employ sophisticated secretion systems, such as the Type II Secretion System (T2SS) and Type III Secretion System (T3SS), to deliver effectors to fight plant cells. These effectors can suppress plant immunity or manipulate host cell processes to favor infection. However, plants have evolved to detect these effectors through specialized resistance proteins, which trigger strong immune responses that halt bacterial proliferation and infection. This ongoing molecular arms race between bacterial effectors and plant defenses underscores the complexity of plant-microbe interactions. Created with BioRender.com.

*Xanthomonas campestris* pv. *campestris*, the causal agent of black rot in crucifers, employs such strategies to colonize the vascular tissues of plants, leading to significant yield losses in crops like cabbage and cauliflower (*Sun et al., 2006*). Similarly, *Xanthomonas oryzae* pv. *oryzae*, which causes bacterial blight in rice, utilizes flagellar motility and effector proteins to invade and spread within rice tissues, resulting in severe disease outbreaks and substantial reductions in rice production (*Niño-liu, Ronald & Bogdanove, 2006*).

Recent advancements in molecular biology have provided deeper insights into the intricate mechanisms underlying flagellin polymorphisms in *Xanthomonas* spp., unraveling their profound implications for host-pathogen interactions. Through comparative genomic analyses, researchers have uncovered a rich diversity of *fliC* alleles within *Xanthomonas* genomes, with each allele often associated with specific host plants or pathovars (*Jacques et al., 2016*). This genomic diversity reflects the evolutionary adaptation

of *Xanthomonas* populations to their respective ecological niches, where interactions with diverse hosts drive the selection of flagellar variants optimized for efficient colonization and infection.

Functional studies have complemented genomic analyses by elucidating the functional consequences of flagellin polymorphisms on bacterial behavior and pathogenicity (*Felix et al., 1999*; *Sun et al., 2006*; *Cai et al., 2011*; *Hao et al., 2014*; *Wang et al., 2015*; *Wei et al., 2020*). Investigations into amino acid substitutions within the flagellin protein have revealed their profound effects on flagellar structure, motility, and immune recognition (*Gottig et al., 2009*; *Robatzek & Wirthmueller, 2013*). Certain amino acid changes can influence flagellar morphology, altering its propulsion capabilities and adhesive properties, thereby impacting the bacterium's ability to move through diverse environments and adhere to host surfaces. Moreover, variations in flagellin structure can directly affect the recognition of *Xanthomonas* by the host immune system, modulating the intensity and specificity of immune responses elicited by infected plants.

In citrus crops, *Xanthomonas axonopodis* subsp. *citri*, responsible for citrus canker, demonstrates how flagellin polymorphisms and other virulence factors contribute to disease. This pathogen uses flagellar motility to reach stomatal openings on citrus leaves, where it forms biofilms and secretes effector proteins that facilitate tissue invasion, leading to characteristic lesions and fruit drop (*Graham et al., 2004*).

Finally, recent research has provided further insights into the tangled relationship between flagellin polymorphisms, external factors, and bacterial pathogenicity. Recent studies show the multifaceted connection among flagellin diversity, gene regulatory mechanisms, and bacterial motility, shedding light on the adaptive strategies and virulence determinants utilized by *Xanthomonas* species affecting solanaceous plants. Findings revealed that tomato plants show a weaker oxidative burst response to *Xanthomonas* flgII-28 peptides compared to *Pseudomonas syringae*, with the flgII-28 variant from *X. euvesicatoria* being inactive. Some *Xanthomonas* species can evade FLS2- and FLS3-mediated immunity, although tomato plants can still recognize flgII-28 from other *Xanthomonas* species, reducing their virulence (*Malvino et al., 2021*).

Bacterial motility varies with flagellin variants and environmental conditions. *X. euvesicatoria* and *X. perforans* exhibit greater motility when induced with chemoattractants compared to *X. vesicatoria* and *X. gardneri*. On the other hand, nutrient starvation suppresses motility by affecting flagellum-related gene expression. Differences in motility among strains might result from variations in *fliC* gene expression (*Malvino et al., 2021*).

Interestingly, it was evidenced how past environmental conditions influence future bacterial behavior. *Xanthomonas* isolates maintained consistent motility before and after ultracold storage, indicating a responsive memory. Therefore, future research should consider prior growth conditions. Variability in flg22 and flgII-28 epitope regions affects bacterial motility and pathogen virulence. While flagellin variants impact motility and host immune recognition, external growth factors and gene expression regulation have a more significant effect. Inducing motility gene expression under chemoattractive conditions

could resolve issues in assessing immunity-triggering flagellar sequence variants on bacterial motility (*Malvino et al., 2021*).

As bacteria have evolved polymorphisms in flg22 peptides to evade plant recognition, plants have similarly evolved variants of the FLS2 receptor to detect these polymorphic peptides. For instance, *Glycine max* (soybean) possesses GmFLS2 homologs that can recognize the flg22 epitope from *Ralstonia solanacearum*, which most other plants cannot detect. Additionally, *Vitis riparia* (riverbank grape) has developed an FLS2XL receptor that can perceive flg22 from *Agrobacterium tumefaciens*, which also evades recognition by most plants Interestingly, in both cases, these receptors have acquired the ability to recognize evading flg22 variants while still detecting canonical flg22 epitopes. The genomes of both plants contain multiple copies of FLS2 homologs, potentially allowing for the evolution of expanded recognition specificity (*Sanguankiattichai, Buscaill & Preston, 2022*). Further studies are necessary to determine if the evolution of these variant receptors has driven changes in pathogen infection mechanisms to overcome this expanded recognition capability or influenced the composition of the plant microbiota.

The diversity of FLS2 receptor variants that detect polymorphic flg22 elicitors provides a valuable resource for improving plant resistance against specific pathogens through the transfer of receptors and co-receptors between plants. For example, transferring GmFLS2 from soybean to tomato confers resistance against *R. solanacearum* (*Wei et al., 2020*), and similarly, transferring FLS2XL to tobacco can limit crown gall disease caused by *A. tumefaciens* (*Tiwari, Mishra & Chakrabarty, 2022*). Additionally, FLS3 can be used to enhance or confer resistance to bacteria that evade FLS2-flg22 recognition.

In parallel, the identification of flagellin polymorphisms as determinants of host specificity and virulence in *Xanthomonas* spp. provides valuable insights for enhancing our understanding and manipulation of host-pathogen interactions in agriculture. By unraveling the molecular basis of flagellar diversity and its ramifications for bacterial pathogenicity, researchers can devise targeted strategies aimed at bolstering plant immunity and mitigating the detrimental effects of bacterial diseases on crop productivity. These comprehensive analyses offer promising avenues for developing disease-resistant crops, as they highlight the potential to leverage flagellin polymorphisms in breeding and biotechnological interventions to enhance plant resilience against bacterial pathogens.

## CONCLUSION

The elaborate interplay between flagella, bacteria, and host plants represents a dynamic and ongoing evolutionary arms race shaped by millions of years of coevolutionary dynamics. Flagella serve as multifunctional organelles that enable bacteria to navigate through complex environments, locate host plants, and initiate interactions leading to colonization and infection. Meanwhile, plants have evolved sophisticated mechanisms to detect and respond to flagellin, a conserved molecular pattern present in bacterial flagella, triggering robust defense responses aimed at limiting bacterial growth and spread.

Recent advances in our understanding of flagellin perception in plants, bacterial motility, and flagellar diversity have provided new insights into the molecular mechanisms underlying host-pathogen interactions. The identification of solanaceous-specific

receptors, such as FLS3, has expanded our knowledge of plant immune recognition systems, while comparative genomic analyses have revealed the importance of flagellin polymorphisms in bacterial adaptation and virulence.

By elucidating the molecular mechanisms governing flagellin perception, bacterial motility, and flagellar diversity, we hope to pave the way for the development of innovative strategies for enhancing plant immunity and mitigating the impact of bacterial diseases on agricultural productivity.

## ACKNOWLEDGEMENTS

I want to express my deepest gratitude to my two main advisors, Dr. Sarah R. Hind and Dr. Steven J. Clough, for their invaluable mentorship, patience, and guidance throughout my years at the University of Illinois at Urbana-Champaign. They have played a crucial role in shaping my academic and professional trajectory and I am truly grateful for their selfless support.

### Funding

The article processing charge was paid by Bayer CropScience. The funders had no role in study design, data collection and analysis, decision to publish, or preparation of the manuscript.

### Grant Disclosures

The following grant information was disclosed by the authors:
Bayer CropScience.

### Competing Interests

The authors declare that they have no competing interests.

### Author Contributions

- Maria L. Malvino conceived and designed the experiments, performed the experiments, analyzed the data, prepared figures and/or tables, authored or reviewed drafts of the article, and approved the final draft.

### Data Availability

This is a literature review.

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
