# Peer review of "Unraveling the dynamics of Xanthomonas’ flagella: insights into host-pathogen interactions"

_PeerJ, doi:10.7717/peerj.18204_

## Round 0.1 · original submission · Major Revisions

Dear authors:

Thank you for submitting your manuscript to PeerJ. Your research topic is not only interesting but also holds significant potential. Two experts in your research areas have reviewed and recommended revisions to your manuscript. While several areas need improvement, I believe in the value of your work and its potential impact. These areas include insufficient innovation, poor language expression, lack of rigorous logic and recent publications, and difficulty supporting the conclusions drawn from experimental evidence. Additionally, there are some oversights, and the English language must be consistent with the journal's requirements. Both reviewers have provided major and minor comments. As you revise your manuscript, I encourage you to consider the following key points:

1. The literature cited is outdated and does not include the latest research progress, particularly the recent findings on the interaction mechanism between flagellin and bacterial migration.

2. The description of the flagella of Xanthomonas in bacterial migration, infection, pathogenicity, and host immune activation is overly general and lacks a systematic summary and specific examples.

3. To enhance the clarity and impact of your findings, including a systematic and visual graph is crucial. This will summarize the functions of Xanthomonas' flagella and provide a clear and concise representation of its association with migration, infection, pathogenicity, and immune activation. Such a graph will significantly enhance the readability and understanding of your work.

4. The review seems to be missing some essential points. Firstly, it should discuss the known tertiary structure of FLS2 and flagellin and how this knowledge can be utilized to develop new FLS2 receptors with improved flagella detection capabilities. Additionally, it should provide more information and citations on this topic.

5. Furthermore, the review appears that the abstract may be misleading in some areas. For example, the abstract states, “Through a comprehensive analysis of flagellin polymorphisms within the genus Xanthomonas, we elucidate their potential impact on host recognition and bacterial pathogenicity, offering insights into strategies for developing disease-resistant crops.” However, this point is not sufficiently covered in the review.

Best regards,

Sincerely,

Tika Adhikari

Reviewer 1 ·

Basic reporting

1.The cited literature is too outdated and lacks the latest research progress, especially those recent findings on revealing the interaction mechanism between flagellin and bacterial migration.
2.The description of the flagella of Xanthomonas in bacterial migration, infection, pathogenicity, and host immune activation is too general, lacking a systematic summary and specific examples.
3. The entire text lacks a systematic and visual graph to summarize and display the functions of Xanthomonas’ flagella and its association with migration, infection, pathogenicity, and immune activation.

Experimental design

1.The title “Unraveling the Dynamics of Xanthomonas’ Flagella: Insights into Host-Pathogen Interactions” is an attractive review topic, but the content of the main text is not enough to support such a theme, and the author needs to summarize more related literature and recent progress.

Validity of the findings

No comment.

Additional comments

The author has chosen an attractive topic, and the writing of the entire article is relatively standardized and fluent. If the author can add some latest research progress, enrich the content of the flagella of Xanthomonas in terms of pathogen pathogenicity and host immune interaction, and display their interaction mechanisms and coevolutionary relationships through figures or charts, it will be of great benefit to scholars or students related to this field.

Reviewer 2 ·

Basic reporting

This contribution briefly summarizes some fascinating and important research that has been done in recent years. The brevity is a strength – it offers a quick entry point into the topic of flagellin detection by plants and the conflicting need in the bacterial pathogen to have flagella for motility but have them not serve as an elicitor of innate immunity. An exciting focal point for future research is raised (designing/modifying plant immune receptors to overcome flagellin structure variation). I share specific concerns, a few of which are substantial, in part 3 of this referee report. But overall this manuscript is getting close to being an excellent contribution.

Experimental design

As a review article, this seemed to be an odd component of the manuscript and/or the review process and/or journal requirements.
Lines 79-93 “Survey Methodology”
Is this section required by this journal? If not, can (preferably) be deleted.

Validity of the findings

As noted above, this is a promising manuscript that, with a few revisions, can make a substantive contribution.

One angle entirely missing from the review is the known tertiary structure of FLS2 and flagellin, which can play a major role in designing novel FLS2 receptors with revised flagella detection capacity in the future. Some of that work has already been done but there is more to do, and similar work could be done with FLS3, etc. The review would benefit from at least a few new sentences and citations on this aspect of the reviewed topic.

Review is also missing something at/after line 212, or Abstract is misleading in Lines 26-29:
“Through a comprehensive analysis of flagellin polymorphisms within the genus Xanthomonas, we elucidate their potential impact on host recognition and bacterial pathogenicity, offering insights into strategies for developing disease-resistant crops.”
Abstract suggests that this has been done. Text on lines 239-244 seems to suggest that this is a future goal, not an accomplished task. Please clarify (reword abstract or reword sentences and/or add sentences and/or citations in lines 212-244).
The sentence quoted above is potentially the most interesting and novel part of the entire review, so the above deserves particular attention in an improved version of this review, which is otherwise a concise and useful contribution.

Additional specific suggestions for revision include the following:
The term “intricate interplay” is used far too many times in various places. Please find other ways of saying this.

Are there just one or two articles that can be cited about motility of other plant pathogenic bacteria, such as Ralstonia or Pseudomonas?

Lines 161-165: The citations on line 166 belong on line 163. Need to look further for a few papers that support the second sentence, about the conservation of FLS2 across broad species.

It might be interesting to note somewhere that mammalian TLR5 receptors also recognize bacterial flagella and play a significant role in innate immunity to bacteria, but recognize at a different conserved site/epitope of the flagella protein.

Line 203: Sun 2006 should again be cited in support of this sentence, or the sentence ending on line 207.

Line 220-221: This sentence (or with a new sentence elsewhere in the paragraph) screams for one or a few citations about studies of bacteria outside of plant pathogens, as some outstanding work on flagellin structure/function and the capacity to change innate immunity-inducing epitopes has been conducted.

Line 223: Is this Lee 2014 paper the correct citation?

Lines 233-237: Repetitive with prior sentences.
Lots of fancy words appear, but half of the substance of the Malvino 2021 paper is not conveyed. The bacteria’s recent history of growth conditions played a significant role in motility phenotypes, impacting the capacity to assess the roles of immunity-triggering flagellar sequence variants on flagellar function in bacterial motility. The issue could be resolved by inducing motility gene expression first, using chemoattractive conditions.

Line 262-264: Delete this sentence. It states the obvious in a way that detracts from the professionalism of the contribution.


Minor:
Some of the language may be a bit too floral or decorated. Examples:
Lines 74-77, lines 233-237.

Could drop words:
“manuscript” (line 17), “esteemed” (line 30), second “intricate” (line 35), third “intricate” (line 64), “will” (line 71),

Line 219 paragraph break? Or no paragraph break?

Additional comments

We look forward to reading the final revised published version.

---

## Round 0.2 · Minor Revisions

Dear authors,

Thank you for submitting your revised manuscript to PeerJ for publication. In view of reviewers’ comments and my own assessment, there are still some minor issues that need to address before accepting it.

I would appreciate it if you could review both reviews and revise and submit your manuscript.

Thank you.

Tika Adhikari

Reviewer 1 ·

Basic reporting

no comment

Experimental design

no comment

Validity of the findings

no comment

Additional comments

As the authors has answered all the comments point-by-point and made correspondding revision, the whole manuscript has been significantly improved and enhanced. If possible, there is still one point that can be improved, which is that the Figure 1 do not systematically present the full picture of this review, and the figure legend should be described more detailed. Besides, immuno signalling is not widely used, immune signalling maybe better.

Reviewer 2 ·

Basic reporting

Much improved. See my summary comments below.

Experimental design

N/A

Validity of the findings

Valid review of the literature.

Additional comments

The present revised version of this review article is much improved. The author has responded to referee suggestions and successfully made a large number of additions. As with most review articles, one could quibble with any paragraph or section, that some past work is not included or certain ideas receive more emphasis than others, but it is the prerogative of the author to direct readers toward what the author wishes to emphasize. I did not find any paragraphs that seemed unfair or inaccurate, and the overall review now seems to achieve the author's holistic purpose. Any remaining issues with this submission might more fairly be classified as "missed opportunities." But for the purposes of a PeerJ review article, this seems ready to be released for public educational use. It will inspire some scientists to gain interest in the topic and possibly add to it in the future.

---

## Round 0.3 · accepted · Accept

Dear authors,

Thank you for submitting your manuscript to PeerJ for publication. I am pleased to accept it.

Best regards,

Tika Adhikari